# An Economic Dilemma between Molecular Weapon Systems May Explain an Arachno-Atypical Venom in Wasp Spiders (*Argiope bruennichi*)

**DOI:** 10.3390/biom10070978

**Published:** 2020-06-30

**Authors:** Tim Lüddecke, Björn M. von Reumont, Frank Förster, André Billion, Thomas Timm, Günter Lochnit, Andreas Vilcinskas, Sarah Lemke

**Affiliations:** 1Department of Bioresources, Fraunhofer Institute for Molecular Biology and Applied Ecology, Ohlebergsweg 12, 35392 Gießen, Germany; andre.billion@ime.fraunhofer.de (A.B.); andreas.vilcinskas@ime.fraunhofer.de (A.V.); 2LOEWE Centre for Translational Biodiversity Genomics (LOEWE-TBG), Senckenberganlage 25, 60325 Frankfurt, Germany; bmvr@arcor.de (B.M.v.R.); sarah.lemke@ime.fraunhofer.de (S.L.); 3Institute for Insect Biotechnology, Justus-Liebig-University of Gießen, Heinrich-Buff-Ring 26-32, 35392 Gießen, Germany; 4Institute for Bioinformatics and Systems Biology, Justus-Liebig-University of Gießen, Heinrich-Buff-Ring 58, 35392 Gießen, Germany; frank.foerster@computational.bio.uni-giessen.de; 5Institute of Biochemistry, Justus-Liebig-University of Gießen, Friedrichstr. 24, 35392 Gießen, Germany; thomas.timm@biochemie.med.uni-giessen.de (T.T.); guenter.lochnit@biochemie.med.uni-giessen.de (G.L.)

**Keywords:** venomics, *Argiope bruennichi*, CAP superfamily, ICK, neuropeptides, hunting behavior, spider venom, proteotranscriptomics, bioresources

## Abstract

Spiders use venom to subdue their prey, but little is known about the diversity of venoms in different spider families. Given the limited data available for orb-weaver spiders (Araneidae), we selected the wasp spider *Argiope bruennichi* for detailed analysis. Our strategy combined a transcriptomics pipeline based on multiple assemblies with a dual proteomics workflow involving parallel mass spectrometry techniques and electrophoretic profiling. We found that the remarkably simple venom of *A. bruennichi* has an atypical composition compared to other spider venoms, prominently featuring members of the cysteine-rich secretory protein, antigen 5 and pathogenesis-related protein 1 (CAP) superfamily and other, mostly high-molecular-weight proteins. We also detected a subset of potentially novel toxins similar to neuropeptides. We discuss the potential function of these proteins in the context of the unique hunting behavior of wasp spiders, which rely mostly on silk to trap their prey. We propose that the simplicity of the venom evolved to solve an economic dilemma between two competing yet metabolically expensive weapon systems. This study emphasizes the importance of cutting-edge methods to encompass the lineages of smaller venomous species that have yet to be characterized in detail, allowing us to understand the biology of their venom systems and to mine this prolific resource for translational research.

## 1. Introduction

Spiders are diverse and species-rich arthropods that have conquered most terrestrial habitats. The 48,463 extant spider species [1] share a common body plan that has changed little during ~380 million years of evolution. Spiders also possess a unique biochemical toolbox that uses a combination of venom and silk to subdue prey, contributing to their evolutionary success [2]. Moreover, they represent one of the few orders of terrestrial animals in which almost all extant species feature a functional venom system and are thus considered as the most successful group of venomous animals [3]. Accordingly, spiders play a pivotal ecological role as venomous predators by maintaining the equilibrium of insect populations [4].

Venoms are complex mixtures of low-molecular-weight compounds, peptides and proteins, which act as toxins by disrupting important physiological processes when injected into prey [5,6]. They are used for defense, predation or competitor deterrence, but in all cases, they are physiologically expensive traits that have been optimized by strong selective pressure for specific functions. This evolutionary streamlining often results in high selectivity and target-specific bioactivity, meaning that animal venoms are now considered valuable bioresources in the field of drug discovery [7]. Several blockbuster drugs have been derived from venom components [7], but they were also investigated as research tools, cosmetics, industrial enzymes or bioinsecticides [8,9,10,11]. 

Spider venoms tend to be chemically more complex than other animal venoms, and up to 3000 different venom components can be present in a single species [12,13]. It has been estimated that the sum of all spider venoms could ultimately yield 10 million bioactive molecules, but only 0.02% of this diversity has been discovered thus far [14,15]. Several promising drug candidates for stroke, pain, cancer and neural disorders have been identified in spider venoms [16,17,18,19]. The major components of these spider venoms are low-molecular-weight inhibitor cysteine knot (ICK) peptides with robust tertiary structures conferred by the presence of a pseudoknot motif of interweaved disulfide bonds [13,20,21]. Additionally described as knottins, such peptides are often neurotoxic and remarkably resistant to heat, osmotic stress and enzymatic digestion, making them ideal drug candidates [13]. Although ICK peptides are found in other arthropod venoms [22,23,24,25], the diversity of these peptides in spider venoms is unprecedented. Approximately 60% of all spider venom components accessible in UniProt [26] are ICK peptides; hence, these peptides are commonly perceived as the principal component of spider venoms [13].

The analysis of venoms formerly relied on fractionation methods that require large amounts of starting material [27,28]. Therefore, previous studies of spider venoms have focused on species with strong anthropocentric connections, such as those posing direct medical threats or those of extraordinary size, making the venom easier to access in sufficient quantities. This restricted analyses to members of the families Atracidae, Ctenidae, Theraphosidae, Sicariidae and Theridiidae, which represent a narrow sample of spider diversity [1,15]. More recently, the advent of high-throughput methods compatible with miniscule samples has provided the means to expand the scope of such studies from less-accessible species [12,29]. Combinations of genomics, transcriptomics, proteomics and microbiomics [30,31] now allow the analyses of venoms from previously neglected taxa [32] in an emerging field known as modern evolutionary venomics [29].

Several of the most species-rich spider lineages have not been studied at all in the context of venom systems [15,33]. In this study, we therefore considered the family Araneidae (orb weavers), which is the third-largest spider family, comprising 3078 extant species [1]. Orb weavers construct conspicuous and often large orb-like foraging webs, attracting the interest of evolutionary biologists [34,35]. Little is known about their venoms, and only one species (the Chinese orb-weaver *Araneus ventricosus*) has been investigated using a venomics approach [36]. 

We selected the wasp spider *Argiope bruennichi*, which features a wasp-like banding pattern that may have evolved as a tool to lure prey [37]. This species is used as a model organism for the investigation of sexual dimorphism, chemical ecology, reproductive behavior, microbiome analysis and range expansion linked to climate change [38,39,40,41,42,43,44,45,46,47]. Its venom has been extracted for bioactivity assays but has not been analyzed in detail [27]. We applied a cutting-edge proteotranscriptomics workflow, in which an automated multiple-assembly strategy was used as a first step to identify and annotate venom gland-specific transcripts. These were matched to proteome components detected directly using two parallel mass spectrometry (MS) platforms to achieve exhaustive and sensitive protein detection and identification. We discuss the functional components of the venom in the context of wasp spider hunting behavior, in which silk rather than a venomous bite is the primary weapon used to overpower prey [48].

## 2. Materials and Methods 

### 2.1. Collection of Specimens and Sample Preparation for Transcriptomics and Proteomics

Fourteen *A. bruennichi* adult females were collected in September 2018 in Gießen, Germany (N 50.5729555°, E 8.7280508°). Initially, we tried to milk venom from the collected spiders by applying electrostimulation. However, this approach failed due to the small size of the venom apparatus and the low venom yield. Therefore, the strategy was changed, and four days after electrostimulation, whole venom glands were dissected from CO_2_-anesthetized specimens under a stereomicroscope, washed in distilled water and submerged in phosphate-buffered saline (PBS). Venom was released by gentle compression with forceps, and the extracts were centrifuged (10,000× *g*, 10 min, room temperature) to pellet cell debris, before pooling the supernatants for lyophilization. The remaining venom gland tissue was transferred into 1-mL RNAlater solution, pooled and stored at −80 °C. Remaining body tissue was processed in the same manner.

### 2.2. Proteotranscriptomics Overall Workflow

RNA-Seq was used to identify transcripts from venom glands and remaining body tissues, followed by assembly and automated annotation. Crude venom was analyzed by one-dimensional (1D) and two-dimensional (2D) polyacrylamide electrophoresis (PAGE) before two parallel bottom-up MS methods were used to identify the venom proteins. The first was matrix-assisted laser desorption ionization time-of-flight (MALDI-TOF)-MS to characterize peptides derived from the 2D gels, and the second was liquid chromatography electrospray ionization (nanoLC-ESI)-MS to characterize peptides directly from the crude venom samples. Transcripts matching the proteins identified by MS were then analyzed in more detail by examining differential expressions and annotations. The whole workflow in the present analysis was designed to minimize the effects that were described in recent studies, which revealed that transcriptome-based approaches led to the generation of large quantities of false-positive data points and, thus, caused an overestimation of toxic diversity [49]. Strict filter steps were applied—first, only transcripts were included that were two-fold enriched compared to the body tissue. Secondly, transcripts encoding for the vast majority of nonvenom transcripts were excluded from the analysis. On another level, the applied different proteomics platforms were combined to identify the presence of venom transcripts on the protein level. Therefore, our subsequent analysis only considered transcripts encoding for proteomically detectable venom proteins as the baseline in a rather conservative analysis approach that prevented an overinterpretation of the transcriptome data (Figure 1).

### 2.3. Transcriptomics of Venom Gland and Body Tissue

#### 2.3.1. RNA Extraction and Sequencing

RNA extraction and sequencing were outsourced to Macrogen (Seoul, Korea). Following RNA extraction, libraries were constructed using the TruSeq RNA Sample Prep Kit v2 (paired-end, 151-bp read length). Quality was controlled by the verification of PCR-enriched fragment sizes on the Agilent Technologies 2100 Bioanalyzer with the DNA 1000 chip. The library quantity was determined by qPCR using the rapid library standard quantification solution and calculator (Roche). The libraries were sequenced on the Illumina Hi-Seq platform.

#### 2.3.2. Transcriptome Assembly, Annotation and Quantification

Transcriptome data were processed using a modified version of our in-house assembly and annotation pipeline featuring different docker containers for enhanced reproducibility [25]. All containers (Appendix A) were established using Nextflow v19.01.0 (https://www.nextflow.io/). Briefly, all input sequences were inspected using FastQC v0.11.7 before trimming in Trimmomatic v0.38 [50,51] using the settings 2:30:10, LEADING:5, TRAILING:5, SLIDINGWINDOW:4:15 and MINLEN:75. The trimmed reads were corrected using Rcorrector v1.0.3.1 and assembled de novo using a pipeline incorporating Trinity v2.8.4 and rnaSPAdes v3.12 with and without error correction [52,53,54,55]. All contigs were combined into a single assembly, in which transcripts from all assemblers were merged if they were identical. The reads were remapped to the assembly using Hisat2 v2.1.0, and expression values (transcripts per million, TPM) were calculated using stringtie v1.3.5 [56,57,58]. SAM and BAM files were converted using Samtools v1.9 [59]. Open reading frames were then predicted with Transdecoder v5.0.2 [55] and annotated at the amino acid level using Interproscan v5.35-74 and BLASTX v2.6.0+ [60,61] searches against the Swissprot, Toxprot and Arachnoserver databases [14,26]. The resulting assembly was used as a species-specific database for the identification of proteins detected by MS. Sequencing raw data are available at the SRA database (PRJNA634567).

To avoid the overinterpretation of our data, a differential expression analysis was applied to the two samples (venom gland versus remaining body tissue), and only putative venom components derived from transcripts with a logFC > 2 within the venom gland dataset were considered further. Filtering steps were performed within the TBro v1.1.1 framework [62].

### 2.4. Venom Proteomics

#### 2.4.1. Fractionation of Venom Proteins by PAGE

For 1D-PAGE, venom was mixed with tricine sample buffer (Bio-Rad) to make a total volume of 12 μL and incubated for 5 min at 95 °C. The sample was then loaded onto a 16.5% Mini-PROTEAN Tris-Tricine gel (Bio-Rad) in a Mini-PROTEAN Tetra System chamber (Bio-Rad) using 10x Tris-Tricine/SDS running buffer (Bio-Rad). Electrophoresis was carried out at 100 V for 100 min, and protein bands were detected with Flamingo stain (Bio-Rad).

For 2D-PAGE, contaminants were removed from the venom extract by the precipitation of 200-µg protein with 1:4 (*v/v*) chloroform/methanol [63]. The protein pellet was redissolved in 260-μL lysis buffer (6-M urea, 2-M thiourea, 4% CHAPS, 30-mM DTT and GE Healthcare 2% IPG buffer pH 3–10). GE Healthcare IEF strips (pH 3–10NL, 13 cm) were loaded with the sample by rehydration for 22 h, and isoelectric focusing was carried out at gradients of 0–100 V/1 mA/2 W for 5 h, 100–3500 V/ 2 mA/5 W for 6 h and 3500 V/2 mA/5 W for 6 h using a Multiphor II system (GE Healthcare). The IEF strip was then equilibrated for 15 min in 5-mL equilibration stock solution (6-M urea, 4% SDS, 0.01% bromophenol blue, 50-mM Tris-HCl pH 8.8 and 30% (*v/v*) glycerol) containing 65-mM DTT and, then, for 15 min in the same solution containing 200-mM iodacetamide. Proteins were separated in the second dimension on a 14% SDS polyacrylamide gel [64] in a Hoefer600 cell (GE Healthcare) for 15 min at 15 mA (100 V/15 W limits) and 4 h at 150 mA (400 V/60 W limits). The proteins were detected with Flamingo stain (Bio-Rad).

#### 2.4.2. MALDI-TOF-MS

The 2D gel was analyzed using PDQuest (Bio-Rad, CA, USA), and 152 spots (Appendix A) were excised using the ExQuest Spot Cutter (Bio-Rad) and transferred into 96-well plates (Greiner Bio-One). The samples were digested simultaneously by using a MicroStarlet pipetting robot (Hamilton Robotics, NV, USA) to execute the following steps: the excised gel plugs were destained with 25-mM ammonium hydrogen carbonate containing 50% (*v/v*) acetonitrile, dehydrated with 100% acetonitrile, rehydrated in 50-mM ammonium hydrogen carbonate, dehydrated with 100% acetonitrile, dried at 56 °C, rehydrated with 17-μL 25-mM ammonium hydrogen carbonate containing 4.5-ng/µL sequencing grade trypsin (Promega) and 0.025% Proteasemax (Promega) and incubated at 45 °C for 2 h. Peptides were recovered by extraction with 15-μL 1% trifluoroacetic acid (Applied Biosystems) and stored at 4 °C.

MALDI-TOF-MS was performed on an Ultraflex I TOF/TOF mass spectrometer (Bruker Daltonics) equipped with a nitrogen laser and a LIFT-MS/MS facility. Summed spectra consisting of 200–400 individual spectra were acquired in positive ion reflectron mode using 5-mg/mL 2,5-dihydroxybenzoic acid (Sigma-Aldrich) and 5-mg/mL methylendiphosphonic acid (Fluka) in 0.1% trifluoroacetic acid as the matrix. For data processing and instrument control, we used the Compass v1.4 software package consisting of FlexControl v3.4, FlexAnalysis v3.4 and BioTools v3.2. Data storage and database searches were carried out using ProteinScape v3.1 (Bruker Daltonics, MA, USA). Proteins were identified by Mascot v2.6.2 (Matrix Science, United Kingdom) peptide mass fingerprinting using the venom gland transcriptome as a database. The search was restricted to peptides larger than 10 amino acids with a mass tolerance of 75 ppm. Carbamidomethylation of cysteine was considered as a global modification, the oxidation of methionine was considered as a variable modification and one missed cleavage site was allowed. Only peptides with a Mascot Score > 80 were considered for further analysis (Appendix A). The proteomic raw data are available at PRIDE (PXD018693).

#### 2.4.3. NanoLC-ESI-MS

We dissolved 10 µg of protein in 25-mM ammonium bicarbonate containing 0.6-nM ProteasMax^TM^. Cysteines were reduced with 5-mM DTT for 30 min at 50 °C and modified with 10-mM iodacetamide for 30 min at 24 °C. The reaction was quenched with an excess of cysteine, and the protein was digested with trypsin at a 50:1 ratio for 16 h at 37 °C. The reaction was stopped by addition trifluoroacetic acid to a final concentration of 1%. The sample was then purified using a C18-ZipTip (Millipore), dried under vacuum and redissolved in 10-µL 0.1% trifluoroacetic acid. 

For analysis, 1 µg of the sample was loaded onto a 50-cm µPAC C18 column (Pharma Fluidics) in 0.1% formic acid at 35 °C. Peptides were eluted with a 3–44% linear gradient of acetonitrile over 240 min, followed by washing with 72% acetonitrile at a constant flow rate of 300 nl/min using a Thermo Fisher Scientific UltiMate 3000RSLCnano device (MA, USA). Eluted samples were injected into an Orbitrap Eclipse Tribrid MS (Thermo Fisher Scientific, MA, USA) in positive ionization mode via an Advion TriVersa NanoMate (Advion BioSciences, NY, USA) with a spray voltage of 1.5 kV and a source temperature at 250 °C. Using the data-independent acquisition mode, full MS scans were acquired every 3 s over a mass range of *m/z* 375–1500 with a resolution of 120,000 and auto-gain control (AGC) set to standard with a maximum injection time of 50 ms. In each cycle, the most intense ions (charge states 2–7) above a threshold ion count of 50,000 were selected with an isolation window of 1.6 *m/z* for higher-energy collisional (HCD) dissociation at a normalized collision energy of 30%. Fragment ion spectra were acquired in the linear ion trap with the scan rate set to rapid, a normal mass range and a maximum injection time of 100 ms. Following fragmentation, selected precursor ions were excluded for 15 s.

Data were acquired with Xcalibur v4.3.73.11. (Thermo Fisher Scientific, MA, USA) and analyzed using Proteome Discoverer v2.4.0.305 (Thermo Fisher Scientific, MA, USA). Mascot v2.6.2 was used to search against the transcriptome database. A precursor ion mass tolerance of 10 ppm was applied. Carbamidomethylation of cysteine was considered as a global modification, the oxidation of methionine was considered as a variable modification and one missed cleavage site was allowed. Fragment ion mass tolerance was set to 0.8 Da for the linear ion trap MS^2^ detection. The false discovery rate (FDR) for peptide identification was limited to 0.01 using a decoy database. For subsequent analysis, we only considered proteins with a Mascot Score > 30 and at least two verified peptides (Appendix A). The raw proteomic raw data are available at PRIDE (PXD018693).

### 2.5. Reanalysis of Araneus Ventricosus Venom

In order to increase the comparability of our results regarding the general conclusions on venoms of Araneidae, we reanalyzed the available proteomic data for only the in-detail studied araneid taxon *Araneus ventricosus*. Since the original work did not perform any assignments to protein classes, all known proteins of *A. ventricosus* were retrieved from the Arachnoserver database [14] and assigned to protein classes by Interproscan [61] (Appendix A). The resulting sequences and protein diversity were assessed manually; however, expression levels could not be included, because the original work did not include quantitative data.

## 3. Results

### 3.1. The A. bruennichi Venom Gland and Body Tissue Yield High-Quality Transcriptome Libraries

Venom glands were dissected from 14 female *A. bruennichi* specimens, and the venom was extracted and set aside for proteomic analysis. The venom glands and remaining body tissues were separately pooled, and RNA was extracted for RNA-Seq analysis. The resulting paired-end libraries were checked for DNA quantity and quality. The concentration of the venom gland transcriptome library was 116.26 ng/µL (fragment size = 387 bp), and the concentration of the remaining body tissue library was 91.47 ng/µL (fragment size = 363 bp). The venom gland transcriptome contained a total of 133,263,138 paired-end reads, with a GC content of 42.2%, a Q20 of 98.2% and a Q30 of 94.5%. The remaining body tissue transcriptome contained a total of 145,808,360 paired-end reads, with a GC content of 41.6%, a Q20 of 98.3% and a Q30 of 94.8%. The libraries were sequenced and annotated using our automated pipeline. 

### 3.2. Only Large A. bruennichi Venom Proteins Are Detected by SDS-PAGE and MALDI-TOF-MS

The crude venom set aside prior to RNA extraction was first fractionated by 1D-PAGE. The lanes representing both concentrations showed identical banding patterns after staining, and the vast majority of the protein bands were in the 25–100 kDa range, with a few weaker bands of 15–25 kDa and no prominent bands below 15 kDa (Figure 2). To characterize the properties of these proteins in more detail, the venom sample was fractionated by 2D-PAGE. The isoelectric focusing steps (pH 3–10) revealed proteins with a range of pI values, although predominantly focused around pH 7 (Figure 2). In agreement with the 1D-PAGE results, the orthogonal SDS-PAGE step indicated that most spots represented proteins of 25 kDa or more, with only a few in the size range 10–25 kDa and hardly any below 10 kDa (Figure 2).

We excised 152 spots from the 2D gels for MALDI-TOF-MS analysis, 41 of which matched to significantly enriched predicted coding regions from our venom gland transcriptome, while the remainder matched nonvenom proteins. Among the 41 venom-related spots, only six were ultimately assigned to protein classes known to be present in other animal venoms (Table 1). The sequences of the venom proteins identified in *A. bruennichi* were highly similar to the toxins previously identified in its close relative, the Chinese orb-weaver *A. ventricosus* [36]. Their molecular masses fell with the range 28.3–50.5 kDa, and functional annotation revealed that they all belong to the cysteine-rich secretory protein, antigen 5 and pathogenesis-related protein 1 (CAP) superfamily.

### 3.3. Further A. bruennichi Venom Proteins Are Revealed by High-Resolution NanoLC-ESI-MS

In a parallel proteomics workflow, the crude venom was analyzed by high-resolution nanoLC-ESI-MS (Orbitrap), revealing a total of 1806 protein groups, including 415 predicted coding regions matching significantly enriched predicted coding regions from our venom gland transcriptome. In size, protein groups ranged from 2 kDa to 950 kDa (Figure 3). The majority of the identified protein groups within the wasp spider venom are composed of isoforms between 10 kDa and 100 kDa; only a marginal fraction of the protein groups is sized below 10 kDa, over 100 kDa or even over 200 kDa. 

From identified protein groups, we retrieved 54 protein groups with putative venom functions, representing 20 different protein families. Many of these protein families have previously been identified in spider venoms, including Kunitz-type serine protease inhibitors, prokineticin, EF-hand proteins, MIT-atracotoxins, astacin-like metalloproteases and ICK peptides. Three others showed similarities to hormones and neuropeptides (insulin-like growth factor binding protein (IGFBP), diuretic hormone (DH) and ITG-like peptides). Another class of proteins showed a high sequence similarity to uncharacterized toxins previously isolated from *A. ventricosus* [36]. BLAST searches did not recover any further similar sequences, so the remaining proteins were defined as “unidentified aranetoxins”. The nanoLC-ESI-MS experiment also confirmed our MALDI-TOF-MS data by showing that *A. bruennichi* venom contains multiple CAP proteins that are also the most abundant proteins among the venom components (Table 2).

### 3.4. Data Integration Reveals That A. bruennichi Venom Is Atypical for Spiders

The transcriptomic and proteomic data were integrated for the comprehensive analysis of venom composition in terms of the diversity and abundance (TPM) of venom proteins (Appendix A). In terms of overall diversity, the CAP superfamily was the most represented, with 15 different CAP proteins accounting for more than 27% of all the identified protein components. Leucine-rich repeat proteins and unclassified aranetoxins were also well represented, each with five members, accounting for ~9% of the total diversity. We identified four putative chitinases and serine proteases, each accounting for ~7.5% of the total diversity, and three ICKs and astacin-like metalloproteases, each accounting for ~5.5% of the total diversity. There were two members of the MIT-atracotoxin and ITG-like peptide families, each contributing ~3.5% of the total diversity. Finally, the Kunitz serine protease inhibitor, cystatin, diuretic hormone-like peptide, techylectin, IGFBP, venom protein 11, 5ʹ nucleotidase, prokineticin, thyroglobulin, S10 peptidase and EF-hand families were represented by one member, each contributing <2% to the total diversity (Figure 4). In terms of abundance, CAPs represented 64.3% of the total protein content of *A*. *bruennichi* venom and were by far the most dominant component, followed by ITG-like peptides (9.5%), unclassified aranetoxins (7.7%) and leucine-rich repeat proteins (7.7%). The other components were expressed at much lower levels, with ICKs contributing only 3.3% of the total protein content, followed by putative chitinases (2.6%) and serine proteases (2.7%) and the others each contributing < 1% (Figure 4).

In agreement with the 1D/2D-PAGE experiments, higher-molecular-weight proteins accounted for most of the diversity of the *A*. *bruennichi* venom proteome and were also the most abundant components. We identified 23 proteins/peptides with molecular masses < 20 kDa, but these accounted for only ~42% of the proteome diversity and only ~13% of the total protein content (Figure 4).

### 3.5. The Venom of Araneus Ventricosus Has Similarity to Argiope bruennichi Venom 

The reanalysis of the *A. ventricosus* venom recovered 61 proteins that could be assigned to known protein families, which resembled only a third of its venom components. The remaining uncharacterized proteins mostly comprised short peptides that were currently assigned as diverse but unknown U-aranetoxins, some of these having counterparts in *A. bruennichi*. However, without any available functional information, they cannot be further discussed. 

Proteins of known families include Kunitz serine protease inhibitors (1.6%); MIT-atracotoxins (3.2%); serine proteases (3.2%); ICKs (CSTX, conotoxin and huwentoxin-1 families) (14.5%); thyroglobulins (46.7%) and CAPs (22.6%). Albeit thyroglobulins represent the most diverse protein class in *A. ventricosus* venom and several of the components identified in *A. bruennichi* venom are absent, both venoms share some similarities. Most identified proteins reflect a rather similar venom diversity (Figure 5). This includes CAP proteins identified in *A. ventricosus*, although they are the second-most diverse proteins in its venom. Similar to *A. bruennichi*, many of the identified proteins from *A. ventricosus* represent rather large classes, exceeding 20 kDa. 

## 4. Discussion

Spider venoms typically consist of mostly low-molecular-weight peptides, with ICK peptides as the predominant neurotoxic components. For example, ICK peptides represent 93% of the diversity in *Phoneutria nigriventer* venom [65] and 42 of 46 identified venom components in the barychelid *Trittame loki* [66]. In *Cupiennius salei*, short cationic peptides and ICK peptides together comprise 39% of the venom components, whereas larger proteins only contribute 15% to its diversity [67]. The predominance of ICK peptides has also been reported in venom isolated from *Cyriopagopus hainanus* (formerly *Haplopelma hainanum*), *Selenocosmia jiafu, Lycosa singoriensis* and *Pamphobeteus verdolaga* [68,69,70,71]. The general assumption is that ICK peptides are highly diverse components of spider venom, and dozens of different peptides may be present per species [13]. In contrast, we found that ICK peptides were only a minor component of the *A. bruennichi* venom, with only three different peptides identified (~5.5% of the overall diversity) and a low abundance (only 3.3% of the total content based on TPM counts), suggesting a less important role in wasp spider venom compared to other spiders. Instead, we found that CAP superfamily proteins were both the most diverse (15 different members, >27% of the overall diversity) and the most abundant (>64% of the total content based on TPM counts), suggesting these proteins are particularly important for the function of *A. bruennichi* venom. Given that *A. ventricosus* venom also contains several CAP proteins (accounting for 22.6% of the venom) (Figure 5) [36], we speculate that CAP proteins may be generally important for venom functions in orb-weaving spiders. Unfortunately, we cannot compare our results directly to this previous study, because it was based solely on a proteomic analysis and lacked quantitative data. This is, in particular, of importance, as the simplicity of *A. bruennichi* venom has been recovered by our proteotranscriptomics approach on the level of protein abundance (quantified in TPM) (Figure 4). The extent to which our results regarding the wasp spider venom are to generalize for Araneidae should be a subject for future investigations. However, the atypical nature of *A. bruennichi* venom allows us to develop a functional hypothesis based on the ecology of wasp spiders, focusing on the most dominant venom components.

### 4.1. The Importance of CAP Superfamily Proteins in Wasp Spider Venom

The CAP superfamily is one of several protein groups that have undergone convergent recruitment and neofunctionalization in venom systems, and CAP proteins have therefore been isolated from the venoms of snakes, spiders, cone snails, scorpions, fish, cephalopods and a variety of insects [5]. This taxonomic ubiquity reflects the ability of CAP proteins to adopt diverse functions. For example, CAP proteins in snake venoms act as neurotoxins by interacting with ion channels [5,72,73], whereas CAP proteins in the venoms of lampreys, hematophagous insects and ticks are thought to facilitate feeding [5,74,75]. In bees, wasps and ants, CAP proteins are major allergenic components of venom and are therefore associated with inflammation and potentially fatal anaphylaxis [76,77]. CAP proteins have been detected in several spider venoms but, generally, as minor components, and their function is unknown [66,67]. Thus far, the only known spider with a venom dominated by CAP proteins is *A. bruennichi*. 

The CAP proteins in *A. bruennichi* venom, as in other arthropods, are unlikely to act as neurotoxins, because they lack the C-terminal cysteine-rich domain that confers the neurotoxicity of CAP proteins in snake venom [5]. Phylogenetic reconstructions indicate that spider CAP proteins are similar to Tex31, a well-characterized CAP protein from the venom of the cone snail *Conus textile* [5] that has proteolytic activity [78]. This suggests that CAP proteins in *A. bruennichi* venom (and other spider venoms) may support extra-oral digestion, toxin maturation or act as spreading factors to promote the uptake of other venom components. The lack of CAP superfamily neurotoxins in spiders would not be a disadvantage, because the venom contains other neurotoxic components, including ICK peptides, prokineticins and MIT-atracotoxins. Assuming that the newly identified aranetoxins and neuropeptides also act as neurotoxins, wasp spider venom clearly contains an impressive arsenal of bioactive components that may facilitate hunting. Interestingly, an early study on the effects of spider venom against cockroaches and meal beetles demonstrated that wasp spider venom can paralyze but not kill both these prey [27]. Therefore, despite the atypical composition of *A. bruennichi* venom, this species is nevertheless capable of neurotoxic envenomation.

### 4.2. Wasp Spider Venom Contains Potential New Toxin Classes Similar to Arthropod Neuropeptides

Our proteomic analysis of *A. bruennichi* venom identified five polypeptides that we grouped as unclassified aranetoxins, showing a high sequence similarity to the recently discovered U_6_ and U_8_ aranetoxins in *A. ventricosus* [36]. They are not yet formally assigned to any known class of toxins, and their molecular and biological functions remain to be determined. However, the five unclassified aranetoxins were expressed at high levels in our venom gland transcriptome dataset and are therefore likely to fulfill important functions in the *A. bruennichi* venom system. Their presence in two orb weavers but no other spider families suggests their role may be specific to the unique ecological niche of orb weavers. 

We also identified one diuretic hormone-like peptide, one IGFBP and two ITG-like peptides. Diuretic hormone-like peptides contain a DH31-like domain and are related to a diuretic hormone from the Florida carpenter ant (*Camponotus floridanus*) and, to a lesser extent, U-scoloptoxin-Sm2a from the centipede *Scolopendra morsitans* [79]. This class of protein has not been detected in other spider venoms. The *A. bruennichi* IGFBP is closely related to a protein found in the venom of the tiger wandering spider *Cupiennius salei* [80]. Such proteins are commonly found in arachnid venoms but also in scorpions (*Superstitia donensis*, *Hadrurus spadix* and *Centruroides hentzi*) and ticks of the genus *Amblyomma* [81,82,83,84]. A related protein is encoded in the genome of *A. ventricosus,* but its function has not yet been determined [85]. Whereas the *A. bruennichi* diuretic hormone-like peptide and insulin-like growth factor-binding protein are relatively minor venom components, the two ITG-like peptides were expressed at high levels in the venom gland transcriptome and may therefore fulfil more important functions. They are closely related to peptides found in the black cutworm moth (*Agrotis ipsilon*) and *C. floridanus* but have not been identified in other spider venoms. 

The role of all three classes of proteins described above is unclear, but hormones in other venomous animals have been weaponized as toxins to subdue prey. Such neofunctionalization might occur when hormones recruited to the venom gland for normal physiological activity undergo mutations that affect their surface chemistry and potential for functional interactions. For example, a neuropeptide that regulates physiological processes in the predator could become a toxin if a mutation causes it to interact with a different receptor in a prey species following envenomation. If this process occurs in the context of gene duplication and divergence, the new role in envenomation could be unlinked from the original physiological role, allowing evolutionary forces to fix the neuropeptide as a venom toxin. The neofunctionalization of hormones and neuropeptides in venom systems is further highlighted by the recent discovery of the convergent recruitment of hyperglycemic hormones in the venom of spiders and centipedes [79]. This study demonstrated that helical arthropod-neuropeptide-derived (HAND) toxins are derived from hormones of the ion transport peptide/crustacean hyperglycemic hormone (ITP/CHH) family, which are ubiquitous and functionally diverse neuropeptides in arthropods. ITP/CHH peptides have also been recruited into the venom systems of ticks and wasps and are not restricted to the HAND family. For example, emerald jewel wasp (*Ampulex compressa*) venom contains tachykinin and corazonin neuropeptides that induce hypokinesia in cockroaches [86], whereas exendin from the venom of helodermatid lizards is a modified glucagon-like peptide that interferes with pancreatic insulin release [87]. Amphibians have also recruited a variety of hormone peptides as skin toxins [88,89,90,91]. The novel neuropeptides in *A. bruennichi* might fulfill other functions, and their potential role as toxins must be tested, but the strong expression of the ITG-like peptides indicates an important function in the venom system.

### 4.3. The Potential Ecological Role of Atypical Wasp Spider Venom

The atypical venom composition of *A. bruennichi* could be explained by trophic specialization, which would select for simple venoms prioritizing specific components needed to subdue selected prey species. This would contrast with generalist predators, where diverse venom components would confer a selective advantage [92,93,94,95,96]. However, *A. bruennichi* is not regarded as a specialist feeder, and an alternative explanation must be sought [97].

In a pioneering study, the hunting behaviors of three orb weavers (*Nephila claviceps*, *Argiope aurantia* and *Argiope argentata*) were compared using bombardier beetles (*Brachinus* spp.) as prey [48]. These beetles have evolved a unique chemical defense involving the stress-triggered release of phenolic compounds from abdominal glands under high pressure. The phenolic compounds undergo rapid exothermal oxidation to benzoquinones, thus spraying predators with a pressurized discharge at temperatures up to 100 °C, allowing the beetle to escape from most situations [98]. When such beetles were offered to the three spiders, the interactions were distinct: *N. claviceps* always tried to inject venom as a first-attack strategy, and this resulted in a successful defense and escape by the beetles, whereas both *Argiope* species deployed silk as a first-attack strategy, and envenomation would follow only when the beetle was fully covered and unable to move [48]. In another study using *A. bruennichi* as the model predator, silk was deployed to overpower most prey insects, including those with other robust defense systems such as wasps, and only lepidopteran prey were attacked with venom first [99]. Similar findings have been reported for eight other *Argiope* species, suggesting that this specialized hunting behavior is highly conserved within the genus [99,100,101]. The prevalence of this silk-based hunting strategy may help to explain the simplicity of its venom, which would be under selection solely for its ability to subdue lepidopteran prey.

In venomous animals, each toxin is a valuable resource that contributes to its fitness by facilitating predation, but this advantage must be balanced against the metabolic costs of replenishing venom stocks [102,103,104]. Many venomous animals have evolved as trophic specialists to reduce these costs, and some even produce different venoms for different purposes [105]. Spiders face a similar dilemma, because they possess two potentially competing systems to subdue prey, namely their venom and silk glands. In both cases, protein resources are deployed as a means to facilitate predation, and in both cases, the glands must be replenished at a significant metabolic cost [106]. We propose that the simplicity of *A. bruennichi* venom may reflect the evolutionary consequences of the competition for resources between the venom and silk systems, which have driven its behavioral specialization to use different strategies against different prey species. Intriguingly, the “silk-first” strategy provides the wasp spider with the competitive advantage of a high success rate against even well-defended prey [48,99], potentially contributing to its unprecedented success during its recent range expansion [42,43].

## 5. Conclusions

Our detailed analysis of *A. bruennichi* venom identified potential new classes of toxins and potential new roles for known protein families, including the predominant CAP superfamily. A comparison to the venom of *A. ventricosus* revealed also that other araneid spiders harbor many CAP proteins in their venom, and thus, these proteins may represent group-specific key components for orb-weaver venoms. The molecular functions and biological roles of these proteins should be investigated in detail to disentangle the venom biology of wasp spiders and their relatives and to identify new drug leads. The sequences we identified can be used to produce recombinant *A. bruennichi* venom proteins in larger quantities for detailed functional analyses, and that work is already underway in our laboratory. However, wasp spiders are small animals with limited venom yields and are therefore unsuitable for traditional fractionation workflows [29,32]. Our venomics workflow overcomes this issue by combining the data-driven selection of interesting candidates based on venom gland transcriptome analysis with a dual proteomics strategy for the comprehensive identification of venom proteins directly. Importantly, our highly sensitive venomics workflow means that a comprehensive venom composition is possible starting with only 14 spiders. A similar approach was recently used to analyze the venom proteome of the spider family Pholcidae [107]. 

Such novel workflows and technical platforms will help to extend our knowledge of venom compositions beyond the small collection of amenable organisms with readily accessible venom systems. This has already shown that many commonly held assumptions about venom (based on the limited number of investigated species) are not supported when more diverse species are included. We describe the venom of *A. bruennichi* as atypical for spiders, because its composition, dominated by CAP superfamily proteins and with ICK peptides fulfilling a minor role, differs from the restricted range of spider venoms that have been investigated thus far. Similarly, the recent proteomic analysis of pholcid venom also revealed a distinct composition dominated by neprilysin metalloproteases [107]. Both studies highlight the importance of filling the taxonomic gaps in venom research in order to fully understand the hidden molecular diversity. This is likely to reveal there is no “typical” spider venom but, rather, a spectrum of compositions reflecting different ecological niches. Such diversity will not only illuminate the field of arachnid evolutionary biology but will also provide many more promising candidates for translational research.

## Figures and Tables

**Figure 1 biomolecules-10-00978-f001:**
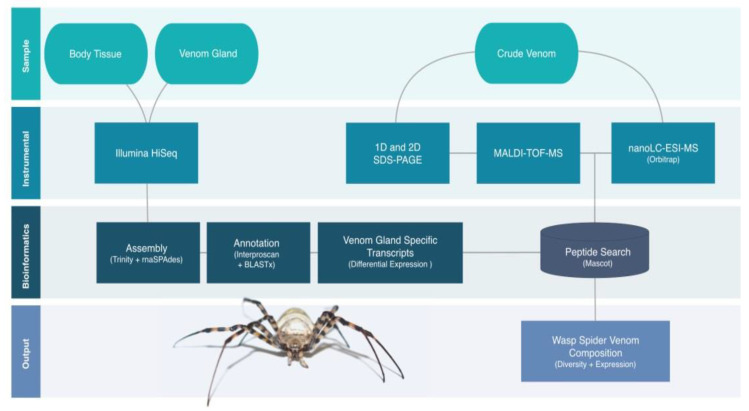
Proteotranscriptomics workflow to characterize the venom of *Argiope bruennichi*. Transcriptomes of venom glands and body tissue were sequenced and assembled. Crude venom was analyzed by 1D/2D-polyacrylamide electrophoresis (PAGE) before combinatorial matrix-assisted laser desorption ionization time-of-flight mass spectrometry (MALDI-TOF-MS) and liquid chromatography electrospray ionization (nanoLC-ESI)-MS. The final transcriptome assembly was used for the MS peptide search. Venom-specific transcripts matching detected proteins were then investigated in terms of expression levels and annotations.

**Figure 2 biomolecules-10-00978-f002:**
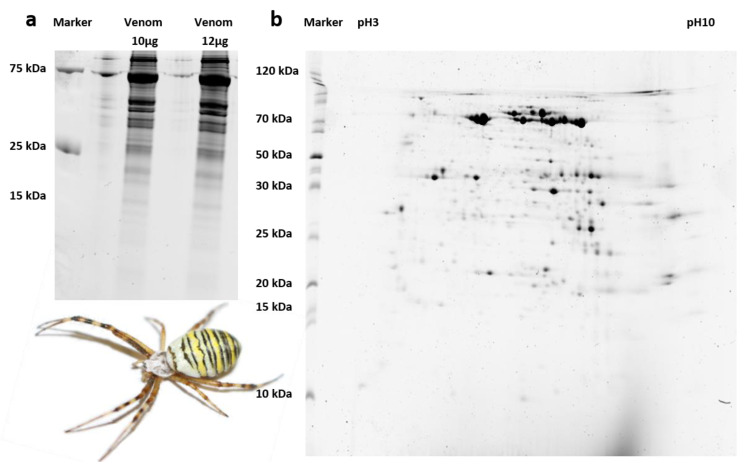
Analysis of *A. bruennichi* venom proteins by PAGE. (**a**) 1D-PAGE of venom proteins at two concentrations, showing identical banding patterns, with most proteins larger than 25 kDa. (**b**) 2D-PAGE, showing that the proteins cover a range of pI values but cluster around pH 7 and confirming that most proteins are larger than 25 kDa.

**Figure 3 biomolecules-10-00978-f003:**
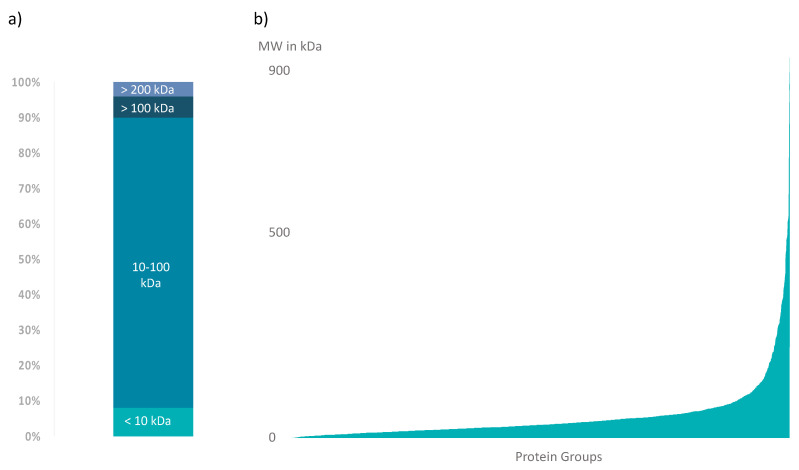
Size distribution within the *A. bruennichi* venom proteome. (**a**) Relative distribution of the identified protein groups within the proteome. Small protein groups of <10 kDa account for 8%, and protein groups between 10 kDa and 100 kDa account for 82%, while protein groups larger than 100 kDa and larger than 200 kDa contribute 6% and 4%, respectively, to the proteomic dataset. (**b**) Absolute size distribution of identified protein groups in kDa.

**Figure 4 biomolecules-10-00978-f004:**
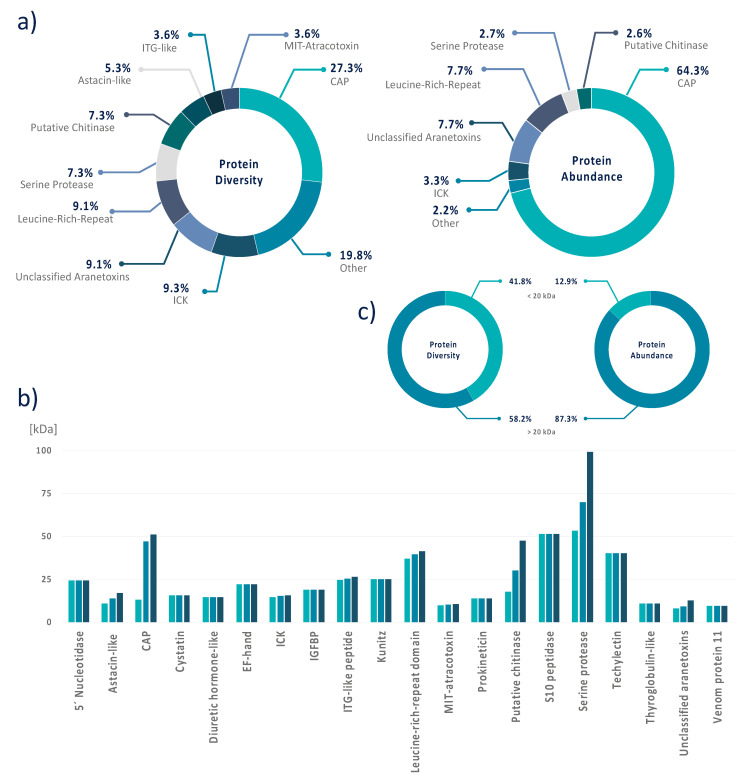
The venom protein profile of *A. bruennichi*. (**a**) Pie charts depict the venom composition in terms of protein diversity based on the number of distinct predicted coding sequences compared to protein abundance based on the transcripts per million reads for each coding sequence. By both measures, CAP family proteins are the dominant venom component, with 15 different members, many expressed at high levels. (**b**) The molecular weight (kDa) of identified venom proteins, with the lowest, average and highest molecular weights per group from left to right. (**c**) The distribution of small (<20 kDa) and large (>20 kDa) proteins in terms of protein diversity and protein abundance (TPM).

**Figure 5 biomolecules-10-00978-f005:**
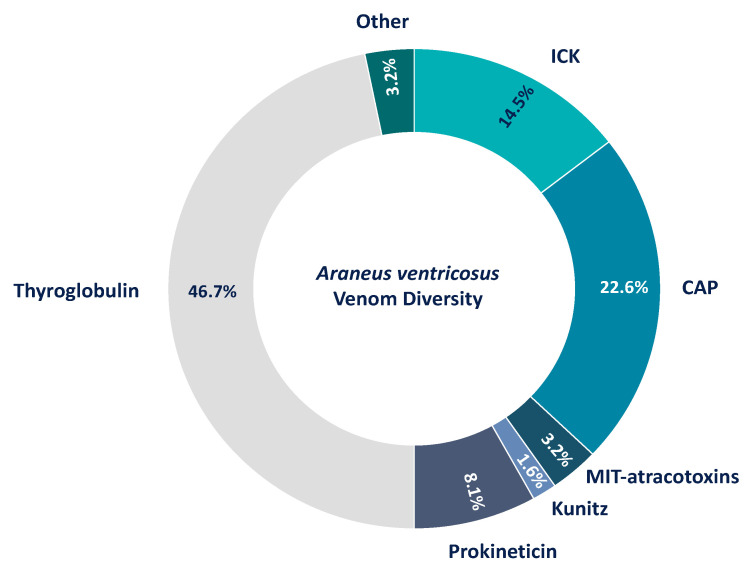
The venom protein profile of *Araneus ventricosus* derived from the reanalysis of the original study [36]. Given are identified proteins that could be annotated via Interproscan in percentages of the total diversity. From the 62 proteins within the dataset, most were assigned as thyroglobulins and, similar to *A. bruennichi*, CAPs.

**Table 1 biomolecules-10-00978-t001:** Identification of *Argiope bruennichi* venom proteins by matrix-assisted laser desorption ionization time-of-flight mass spectrometry (MALDI-TOF-MS). Among 152 spots excised from 2D gels, 41 represented proteins that were enriched in the venom glands, and six of these were similar to previously identified venom components, all putative members of the CAP superfamily.

Spot ID	Class	Score	kDa	Peptides	Coverage (%)	ppm
8501	CAP	136.0	48.9	14	21.1	13.81
7309	CAP	70.7	28.3	6	18.3	7.42
7501	CAP	182.0	50.0	18	35.3	23.55
6502	CAP	85.9	50.5	10	24.8	22.95
6502	CAP	124.0	45.7	15	33.7	23.73
6502	CAP	89.2	48.1	11	25.3	23.57

CAP = cysteine-rich secretory protein, antigen 5 and pathogenesis-related protein 1.

**Table 2 biomolecules-10-00978-t002:** Identification of *A. bruennichi* venom proteins by liquid chromatography electrospray ionization (nanoLC-ESI)-MS. The analysis of peptide fragments allowed us to identify protein groups with putative venom functions, representing 20 different protein families. Confirming the parallel MALDI-TOF-MS analysis, most proteins could be assigned to the CAP superfamily. ICK = inhibitor cysteine knot and IGFBP = insulin-like growth factor binding protein.

Protein Class	Matched Peptides	MW (kDa)	Calc. pI	Mascot Score	Coverage (%)
5′ Nucleotidase	4	24.3	4.54	228	17
Astacin-like metalloprotease	2	10.9	4.79	149	15
Astacin-like metalloprotease	2	14.4	5.87	76	13
Astacin-like metalloprotease	3	17.0	7.93	50	24
CAP	23	51.2	7.77	3697	57
CAP	18	50.8	8.19	2333	52
CAP	14	50.0	7.65	2164	40
CAP	9	28.3	7.66	1918	59
CAP	16	48.9	8.44	1904	46
CAP	8	50.5	7.97	1661	26
CAP	15	51.0	8.50	1127	40
CAP	14	48.0	8.07	1020	46
CAP	11	28.9	9.09	944	44
CAP	5	17.0	5.14	604	32
CAP	6	13.2	8.79	311	49
CAP	3	25.1	6.84	173	27
Cystatin	2	15.6	7.33	48	22
Diuretic hormone-like	2	14.6	9.55	267	23
EF-hand	7	22.3	5.25	112	33
ICK	8	15.0	6.15	1076	40
ICK	2	14.7	4.68	209	14
ICK	4	15.7	6.76	59	24
IGFBP	2	18.8	4.92	63	16
ITG-like peptide	10	26.6	4.78	1666	51
ITG-like peptide	8	24.7	4.96	673	38
Kunitz	2	25.1	7.46	38	8
Leucine-rich-repeat domain	20	39.7	4.93	4279	71
Leucine-rich-repeat domain	10	41.0	5.29	771	34
Leucine-rich-repeat domain	9	36.9	5.74	428	37
Leucine-rich-repeat domain	7	39.2	5.11	215	27
Leucine-rich-repeat domain	5	41.2	5.50	117	23
MIT-atracotoxin	4	10.7	4,94	448	56
MIT-atracotoxin	5	9.8	5.50	140	66
Prokineticin	4	13.8	7.97	636	33
Putative chitinase	12	35.5	7.30	1159	45
Putative chitinase	8	30.1	6.49	219	39
Putative chitinase	2	18.0	5.19	112	11
Putative chitinase	3	47.4	11.15	53	13
S10 peptidase	4	51.4	8.07	118	15
Serine protease	8	53.2	6.40	608	24
Serine protease	8	53.2	6.64	469	23
Serine protease	5	86.2	6.54	135	9
Serine protease	3	99.1	6.27	100	4
Serine protease	3	55.2	6.13	54	7
Techylectin	3	40.3	7.17	88	7
Thyroglobulin-like	3	10.9	7.56	134	31
Unclassified aranetoxins	3	8.3	8.07	630	55
Unclassified aranetoxins	5	12.9	9.57	220	32
Unclassified aranetoxins	4	8.2	8.18	206	39
Unclassified aranetoxins	4	8.3	8.18	202	39
Unclassified aranetoxins	2	8.4	7.71	56	39
Venom protein 11	2	9.5	8.00	234	35

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
