# Peer review of "An Economic Dilemma between Molecular Weapon Systems May Explain an Arachno-Atypical Venom in Wasp Spiders (Argiope bruennichi)"

_biomolecules, 2020, doi:10.3390/biom10070978_

Round 1
Reviewer 1 Report
The study by Lüddecke et al. for the first time provides a detailed analysis of the venom gland transcriptome and proteome of the wasp spider Argiope bruennichi. I quite enjoyed reading the manuscript, which is very well written, making it easy to follow the argumentation provided by the authors. The employed methodology is state-of-the-art and the conclusions provided by the authors are supported by their data. Overall, I would recommend this manuscript for publication in Biomolecules. However, I do have a number of points that should first be clarified during a revision, before the manuscript can be published in Biomolecules.
- L30 “smaller lineages of venomous species” > maybe better “lineages of smaller venomous species”, because there are 3051 species in Araneidae, so it’s actually one of the largest/most diverse spider families
- L40: “Moreover, they represent the only order of terrestrial animals in which almost all extant species feature a functional venom system” > what about scorpions, pseudoscorpions, and the 4 orders of centipedes (I’m not sure myself whether they are almost exclusively venomous or not…)?
- L52: another new reference for toxins as research tools and for biodiscovery would be https://www.sciencedirect.com/science/article/pii/S0006295220303324?via%3Dihub
- L53: even > 3000 venom peptides, see Fig. 1E in Pineda et al, PNAS 2020
- L114 +L280: A. bruennichi
- L178: introduce abbreviation LIFT first
- L189: remove one “raw”
- L191ff: rather than indicating masses of compounds dissolved in certain volumes, the molar ratios should be indicated for each reaction.
- For obtaining RNA from spider venom glands for transcriptomic work, spiders are usually milked and then the venom glands are dissected 3-4 days later. During this time, the spiders will start to regenerate their depleted venom stocks and RNA levels are therefore ramped up. In the current study, it seems that both the venom and the tissue for the RNA extraction were collected at the same time/day. One explanation for the reported “simple” composition of the A. bruennichi transcriptome might be that at this stage, only RNA for peptides and proteins necessary for general maintenance of the venom gland composition was produced. The RNA expression pattern could have been rather different from a depleted venom gland that ramps up toxin production to restore all previously spent/extracted venom components. Can the authors exclude that the timing of RNA collection had any influence on the observed complexity of the transcriptome? This should be discussed.
- Table 2: I would recommend to set the line spacing to 1, to avoid empty lines in the table and to reduce the overall length. If the table cannot fit onto a single page (which would be ideal, maybe the text size might need to be decreased to achieve it), then I would at least show the heading line on the second page as well. The order in which the protein matches are currently presented also seems rather random.
- Fig. 3 a and b: Having the text describing the protein families (e.g. CAP, ITG-like) in light grey and in a small size makes it pretty hard to read. I would increase the size and change to a darker color.
- L326: it would be good to indicate the percentages for diversity of the CAP proteins in A. ventricosus venom, i.e. how many CAP proteins were found based on the total protein/peptide count?
- Please introduce abbreviations on their first use, e.g. IGFBP, ITG, DH,
- L402ff: I agree with the authors that the predominant usage of silk could have resulted in a rather less complex venom composition. On the other hand, the composition of the Latrodectus tredecimguttatus venom transcriptome (see He et al, Plos One 2013) seems to be dominated by large protein toxins and small peptide toxins, despite Latrodectus spec. (like Pholcid spiders) are also being known for using a “silk first” strategy to incapacitate prey, before applying venom.
Author Response
Dear colleague,
thank you very much for revising our manuscript and for giving us such a constructive and really understandable feedback. My coauthors and I basically agreed on all points issued by your review and addressed them throughout the manuscript. Please find a detailed point-by-point reply to all concerns attached.
Sincerely yours,
Tim Lüddecke

Reviewer 2 Report
In their manuscript, Luddecke et al sequenced and assembled transcriptome libraries using dissected Argiope bruennichi venom glands and tissues from the rest of the body. Putative venom toxins were identified based on criteria such as sequence similarity to spider sequences and logFC > 2 (venom glands vs body tissues). These sequences were next used as search database for mass spectra obtained from proteomic studies using extracted venom. Two different proteomic approaches were used: (1) sequencing of digested protein spots from 2D-PAGE; and (2) shotgun-LC-MS/MS. The authors found that, unlike most other reported spider venom, A. bruennichi venom appeared to comprise high molecular weight proteins and certain classes of neuropeptides. The authors theorized that the relatively simplified venom composition of this spider may be a result of balancing the venom and the silk as hunting strategy in hunting of prey.
This is an important study since there are very few detailed analysis of venom of orb weavers available. The manuscript is well-written. I would recommend acceptance of this manuscript for publication in Biomolecules. Here are some comments I hope the authors can consider for submission of a revised manuscript.
- A figure or two to better describe the venom gland library would help the readers have a better gasp of the overall venom compositions. For example, families of toxins identified in venom gland library and relative abundance of their reads, similar to Figure 3A.
- A different figure showing molecular weight distribution (for translated protein sequence) of transcripts sequence may also needed. Addressing Point 1 & 2 would help readers identify if there are toxins families identified in the transcriptome but not in the proteome. As the lack of lower molecular weight proteins/peptides, especially ICK peptides that are so dominant in other known venoms of spiders is an important discussion point, it would be important for the authors to demonstrate that these sequences were not: (1) lost during assembly of transcriptome; and (2) lost during proteomic experiments or analysis. Granted, the use of shotgun-LC-MS/MS should be less susceptible than 2D PAGE to missing of peptides <8 kDa, the complete lack of toxins with MW <8 kDa is curious and should be substantiated with careful analysis. This is especially important since multi-domain ICK & Kunitz-type inhibitors appeared to be present, making it much less likely that single domain ICK & Kunitz peptides would be completely absent.
- Since Araneus ventricosus is the closest relative that has available proteomic information, maybe more comparisons with ventricosus is warranted. For example, a quick look at the A. ventricosus paper appeared to mention high abundant of proteins between 5 to 18kDa, again maybe the authors can discuss their results in this context.
- Other minor comments:
- P2 L74: “from”
- P3 L107: the use of the term intact peptides here can be confused with detection of molecular weights of full-length proteins, although the procedures and results clearly indicate that the authors are talking about digested proteins. The authors should consider using more precise terminologies.
- P5 L191: 0.1% ProteaseMax?
- P10 L282 (and elsewhere, eg Fig 3 legend): Is “abundance of (venom) proteins” determined based on number of read? If that is the case, shouldn’t it be relative level of mRNA expression instead of quantification of the amount of protein? Transcript level may not correlate well with protein expression level, especially when it’s just a snapshot of transcript level. The authors may consider to be more precise in their description.
Round 2
Reviewer 2 Report
I would like to thank the authors for adequately addressing my previous comments. I would recommend acceptance of the manuscript for publication. Congratulations on an excellent study and nicely written article.